

# Incorporating multi-source remote sensing in the detection of earthquake-damaged buildings based on logistic regression modelling

Qiang Li, Jingfa Zhang, Hongbo Jiang

Institute of Crustal Dynamics, China Earthquake Administration, Beijing, No. 1 Anning Zhuang Road, Xisanqi, Haidian District, China, 100085

*Correspondence to*: Qiang Li (liqiang08@163.com)

**Abstract.** After an earthquake, efficiently and accurately acquiring information about damaged buildings can help reduce casualties. Earth observation data have been widely used to map affected areas after earthquakes. However, we can obtain

different modes of remote sensing data from different sensors after an earthquake to assess the damage, manage rescue operations and to estimate economic losses. In this work, for quantification and precision purposes, information on earthquake-induced building damage is extracted using multi-source remote sensing images collected after an earthquake. The multi-source remote sensing data include optical data, synthetic aperture radar (SAR) data, and digital surface model (DSM) data generated by interpolating light detection and ranging (LiDAR) point cloud data. The proposed approach relies

on a pre-existing urban map to identify image objects corresponding to building footprints. The image analysis is carried out according to the rough set theory to further determine the feature parameters with the objective of assessing their effectiveness in singling out changes associated to the building collapse. Features that describe texture, colour, as well geometry are included in our analysis. Logistic regression model was employed to find the optimal fitting function to describe the relationship between the occurrence and absence of destroyed buildings within an individual object. Seven

feature combination models, respectively, based on the use of the texture, colour and geometry, were compared. In our experiment, we considered the whole Beichuan old country comparing classification results with the reference map, which is be regarded as the ground truth for accuracy verification. In our experiment, old Beichuan County, China, the area most devastated by the Wenchuan earthquake on May 12, 2008, is used to test the proposed hypothesis. Through comparison with a ground survey, the experimental results show that the detection accuracy of the proposed method is 94.2%; the area under

the receiver operating characteristic (ROC) curve is 0.827. The efficiency of the proposed method is demonstrated using 6 modes of data combination acquired from the same area. The approach is one of the first attempts to extract damaged buildings through the fusion of three types of data with different features. It addresses multivariate regression methodologies and compares the potentials of different features for application in the field of damage detection.





# 1 Introduction

Earthquake disasters are a type of major natural disaster, and severe earthquakes can cause serious casualties and economic losses through the destruction of a local area. After an earthquake, quickly and accurately obtaining information about the distribution of the disaster is an effective way to reduce losses by improving the rescue work efficiency. Building damage

can reflect the intensity of the ground motion and the economic losses to a certain extent. Quantitative evaluations and high-quality building assessments can also provide important information regarding economic loss, which can be used to allocate resources during restoration and reconstruction. Therefore, quantitatively evaluating building damage due to earthquakes is vital.

Remote sensing technology is an important way to obtain information in the early stage of earthquake relief because of its

objective and efficient access to a wide range of disaster information, which can provide information to support earthquake damage assessment and emergency rescue. There are many works in the literature on this topic that consider both optical and radar data (Voigt et al., 2007; Ehrlich et al., 2009; Corbane et al., 2011). Notably, with the continuous development of satellite technology, the number of satellites operating in orbit and sensors are increasing, and a large amount of remote sensing data can be acquired after a disaster. Most remote sensing data used after a disaster can be classified into three

categories: (1) optical data, (2) synthetic aperture radar (SAR) data, and (3) light detection and ranging (LiDAR) data.

The extraction of information on building damage is the main task of investigations of earthquake damage. Early information extraction mainly used a single data source. However, seismic damage to buildings is complex and variable, and comprehensively identifying seismic information by relying on a single data source is difficult. With the rapid development of space and airborne remote sensing technology, obtaining different types of remote sensing data within a short period after

an earthquake is possible. The effective combination of multi-source data can make full use of the respective advantages of these data to achieve complementary advantages and redundant control. Many researchers have studied building damage information extraction using multi-source remote sensing data. Butenuth et al. (2011) used multi-source remote sensing data to extract road damage information and achieved good results, which indicates its validity in earthquake damage information identification. Chini et al. (2012) investigated the inundation and liquefaction resulting from the 2011 Tohoku tsunami

through a combination of optical data, thematic maps, and SAR data. Stramondo et al. (2006) extracted the distribution of seismic damage from the Izmit and Bam Earthquakes using high-resolution optical images and SAR images. Wang and Jin (2012) proposed a method for identifying seismic damage through a combination of IKONOS, COSMO-SkyMed, and Radarsat-2 images after an earthquake. First, the SAR image before the earthquake was simulated using the geometric information extracted from the optical image. Then, the simulated image was compared with the real image after the

earthquake to extract the building damage information; the extraction accuracy reached 85%. Dell'Acqua et al. (2008) extracted the building damage information of the L'Aquila Earthquake by fusing the optical image and the SAR image; the extraction accuracy was 81%. Based on the GIS information extracted from the optical image and the TerraSAR image after the earthquake, Dong et al. (2011) extracted the Wenchuan earthquake damage information. These authors believed that the




combination of optical and SAR images can effectively improve the extraction accuracy and found that the optical image can be used to identify destroyed buildings. SAR image texture features can be used to identify seismic damage information in blocks. The complexity of the image-forming mechanisms within an earthquake region, especially for radar images, makes the interpretation and analysis of images a challenging task. In particular, discrimination of geometric features is extremely

difficult in the detailed identification of earthquake-damaged buildings.

In addition to the combination of optical and SAR images, the combination of optical and LiDAR images has also made some progress. The integration is mainly based on the texture features of optical data and the elevation information of LiDAR data. Rehor and Voegtle (2008) improved the previous method of extracting buildings by the fusion of optical and LiDAR images. In addition, optimal texture features are used in the calculation. Compared with the use of single optical

images and LiDAR images, the accuracy is improved when these images are combined. Based on the object-oriented support vector machine method, Yu et al. (2011) extracted the information on damaged buildings through the combination of the digital surface model (DSM) obtained from aerial and LiDAR data. LiDAR data and high-resolution images were used to reconstruct a 3D building model; then, the roof of the building model was compared with the pre- and post-earthquake roof patches to quantify the degree of damage to the building. Hussain et al. (2011) detected the ruins of collapsed houses in the

Port-au-Prince area using GeoEye-1 and LiDAR data after the Haiti Earthquake. Through this method, the volume of rubble and debris can be estimated, and the cleaning process can be planned effectively. The features of two types of data are not sufficient to identify different types of buildings. However, there are no data fusion approaches for multi-source remote sensing data in the existing literature.

This paper presents a quantitative evaluation method for seismic damage to buildings based on multi-source remote sensing

data, including optical images, SAR images, and LiDAR images. Different types of images can reflect the different characteristics of intact buildings and collapsed buildings, and the features of seismic damage can be extracted from remote sensing images. Then, the features are reduced according to the selected sample data through rough set theory. Subsequently, the relationship between the degree of damage and the image features is established by the logistic regression method, and a rapid quantitative assessment of the damage degree of buildings is performed.

The main purpose of this paper is to combine multi-source remote sensing data to establish a mathematical model for quantitative evaluation using the logistic statistical analysis method and to achieve a quantitative assessment of earthquake-damaged buildings. Two specific aspects of this work are deemed important for satisfying the final needs of complete and detailed inventories and for ensuring that this work can provide a reliable product (i.e., the damage map, in our case). The first important aspect is a detailed comparison with the rapid assessment of single-image data, which provides a well-known

and reliable decision-making reference. The second aspect is that the work is not just an exercise in collapse detection focused on a few buildings, but rather an evaluation of the capability of multi-source remote sensing in the quantitative assessment of earthquake-damaged buildings, which is important for assessing the role of Earth observations in earthquake damage management with respect to methodological testing on a limited test set.



Specifically, we take old Beichuan County as the study area to demonstrate the effectiveness of the method. Old Beichuan County is one of the areas that was most affected by the Wenchuan earthquake, which led to the relocation of the entire community. Many buildings have been preserved in the form of earthquake memorials. The experiment is based on three different datasets, including Terra-SAR data, post-event airborne optical images, and LiDAR data. The work provides

several steps for addressing the problem, including image processing, feature performance evaluation, and the determination of the advantages of multi-source remote sensing in the damage extraction field. Moreover, this work defines the possible weights and roles of different features for earthquake damage identification, which raises some difficulties when the features are applied to image classification.

This paper introduces the work using multi-source remote sensing images, and the paper is structured as follows. Section 2

introduces the study area and three different datasets. Section 3 describes the characteristics of buildings with different types of earthquake damage in multi-source remote sensing images. The proposed methodologies regarding feature selection, attribute reduction, element determination, and logistic regression model (LRM) are introduced in Section 4. Section 5 presents the results of the experiment and discusses the feasibility of the method. Finally, Section 6 presents the main conclusions.

## 2. Study case and datasets

Old Beichuan County is regarded as the study area in this paper. The Wenchuan earthquake on May 12, 2008, caused a large number of casualties and damage to facilities. Old Beichuan County is one of the areas that was most affected by the Wenchuan earthquake, which led to the relocation of the entire community. Many buildings have been preserved in the form of earthquake memorials. The buildings in the Memorial Park have retained their damaged post-earthquake appearance; thus,

they can be used as the research object of this paper.

The datasets applied in this work include optical multispectral image, LiDAR, and SAR images. In July 2013, the Institute of Crustal Dynamics, China Earthquake Administration, collected a series of unmanned aerial vehicle (UAV) multispectral images from the Wenchuan earthquake site, including old Beichuan County. The resolution of the images is approximately 0.5 m. The optical image we obtained was then rectified. In the same year, we performed an omni-directional scan on the site

of the old town in Beichuan using a ground-based laser scanner. A total of 67 station sites were positioned in old Beichuan County. In 2014, we carried out additional measurements in old Beichuan County. In addition, field investigation data of earthquake damage to buildings were obtained. The SAR images were obtained from the TerraSAR-X data of the German Aerospace Center (DLR). The data are ascending SAR images in the VV polarization mode. The spatial resolution is 1 m.

Combining the careful interpretation of multi-source images and field investigation data, we can obtain a reference map

divided into intact buildings and destroyed buildings. Due to the lack of detailed field survey data, the reference map can be regarded as the ground truth for accuracy verification. The study area location, optical image, and reference map are shown





in Figure 1. Figure 2 presents the multi-source remote sensing images of the study area (Figure 2(a)), SAR image (Figure 2(b)) and DSM image (Figure 2(c)).

## 3. Seismic characteristics of multi-source remote sensing images

The destruction of buildings during an earthquake is complex and diverse. Schweier and Markus (2016) divided the types of
5 building damage into 10 categories, including plane tilt, multi-layer collapse, debris accumulation, and wall tilt. Different sources of remote sensing data reflect different properties. Therefore, the quality of different types of seismic damage in the images also varies.

Generally, the buildings in the optical image were in perfect shape before the earthquake, and the body and shadow of a building are shown as morphological rules and boundaries; the SAR images contain an overlapping area, a reflection area, a
10 roof area, and a shadow area. From the perspective of the characteristics of radar images, intact buildings show a relatively regular arrangement; the spatial relationship between each characteristic is in accordance with the characteristics of the building complex. Due to the characteristics of SAR side-view imaging, in close range, the first characteristic is the overlay area of the wall, which is higher than the surface in the image; hereafter, the wall and surface reflection effect and the dihedral angle of the backscattering intensity on one side of the building are high and show 1- or L-shaped strong echo
characteristics. Moreover, the strong echo location is still consistent with the arrangement of the buildings. In the direction from back to incident, a clear rectangular shadow area can be seen. In the LiDAR images, the three-dimensional structure of a building is clear and complete, and the columns and walls are not missing or tilted, the walls, columns, and ground are at right angles, the wall is smooth and complete, and the point cloud echo is dense.

After a building is destroyed, the square nature of the structure is damaged or disappeared based on the optical images. The
20 boundaries of the building are blurred, or the buildings are level with the surrounding surface. The roof of the building collapsed and broke into pieces. The shadow cannot be clearly identified because of the rubble. The whole building is destroyed, collapsed, or slumped. The original geometric structure in the SAR imagery is blurred or even absent, and the backscattering pattern is changed to multiple scattering in all directions. The main rule of image recognition is that the image brightness is high, and no regular bright spot occurs in a certain position. In the LiDAR image, the three-dimensional shape
of a building is completely absent and in ruins. A large number of peeling walls and reinforcements are present in the middle of and surrounding the building.

Figure 3 illustrates the same intact building obtained from different sensors after the Wenchuan earthquake. In the optical image, the shape is regular, the colour is even, and no abrupt changes occur. From Figure 3(c), we can distinguish the double-bounce line, layover area, and shadow area. The lateral and horizontal roof of the building results in the formation of
30 black areas. In the LiDAR image, the three-dimensional structure is clear, and the wall is smooth and complete.

Figure 4 illustrates the same fully destroyed building obtained from different sensors after the earthquake. In the optical image, the geometry is completely absent, the ground is covered with rubble from the building, and differences in brightness



are present. In the SAR image, the area of the building where the echo signal is poor contains dark colours, overlapping objects, bright lines formed by angle reflections, and shadows and missing features. A small corner reflector is formed locally by the rubble, so there are many highlights. In the LiDAR image, the three-dimensional shape of the building is completely missing and in ruins. A large number of peeling walls and steel can be found in the middle of and surrounding the building.

Figure 3 and Figure 4 show that intact buildings and destroyed buildings have unique features in the optical images, SAR images, and LiDAR images. However, in the optical images, the characteristics of the construction ruins and the ruins of destroyed buildings are similar; additionally, phenomena of "same object with different spectra, different objects with the same spectrum or different optical image spectra, and objects with the same spectrum" occur. The characteristics of large amounts of vegetation in the SAR images appear similar to those of completely destroyed buildings. The method of information extraction based on a single data source can easily cause false alarms. Using traditional methods to achieve high-precision extraction of buildings with different types of seismic damage is difficult. Therefore, the motivation of this paper is to make full use of the characteristics of multi-source remote sensing data and establish a quantitative evaluation model.

## 4. Methodology

Based on the above analysis, buildings with different degrees of earthquake damage can be identified through the comprehensive utilization of multi-source images. Therefore, if the features are selected and a mathematical evaluation model is constructed, then intact buildings and damaged buildings can be quantitatively evaluated. Considering this theory, in this paper, a new approach for detecting earthquake-damaged buildings using post-event multi-source remote sensing images is proposed. A technical flowchart of the proposed methodology is shown in Figure 5.

### 4.1 Data processing

The main inputs of the method are the optical image, SAR image, DSM image, and building sample distribution. Before the feature selection of SAR data, radar sigma naught values can be obtained by formula (1) using the original digital numbers (Infoterra, 2008):

$$\sigma^0 = (k_s \cdot |DN|^2 - NEBN) \cdot \sin \theta_{loc} \qquad (1)$$

where $k_s$ is the calibration and processor scaling factor, DN is the pixel intensity value, and NEBN is the noise equivalent beta naught, and $\theta_{loc}$ is the local incidence angle.



The optical image is a UAV image with a resolution of approximately 0.2 m. The DSM image is generated by the nearest neighbour interpolation method of the ground LiDAR point cloud data collected using the VZ-1000 3D laser scanning system of Austria Riegle Company.

The building distribution map is obtained by carefully interpreting the post-earthquake optical image, LiDAR data, and field investigation data through comparison with the Google image. To carry out this comparative analysis effectively, the optical image, SAR image, and DSM image are corrected using the manual control point selection function implemented into ENVI software. Subsequently, the distribution of buildings can be determined in the three types of images.

## 4.2 Feature selection

Three different types of image features are selected for our experiments, including the spectral features of optical images, the texture features of SAR images and the geometric features of DSM images. The spectral feature is the most intuitive representation of different objects in optical images. The features mainly include maximum, minimum, mean, standard deviation, and brightness. For the detailed calculation process, refer to the relevant reference (Zhao, 2010).

The repeated occurrence of pixel intensity in the spatial position forms the texture of the image. A difference in the spatial arrangement of pixel intensity is expected as a consequence of damage. In particular, features based on second-order statistics are considered in the SAR images obtained from the grey-level co-occurrence matrix (GLCM) method using ENVI software, following the approach proposed by Haralick et al. (1973). Taking four angular directions (0°, 45°, 90°, 135°), a step size d=1 and a window size w=11, four different variables were calculated, and the average was obtained by summing them. From the results of the GLCM, eight texture features, i.e., mean (ME), variance (VA), homogeneity (HOM), contrast (CON), dissimilarity (DIS), entropy (ENT), angular second moment (ASM), and correlation (COR), were chosen for analysis. The features are as follows (Soh and Tsatsoulis, 1999; Wood et al., 2012).

In addition to elevation information, the main feature of the DSM image is the geometric feature. The geometric feature is the spatial distribution of the pixels. The covariance matrix is used to perform statistical analysis (Chen, 2007):

$$S=\begin{bmatrix} Var(X) & Cov(XY) \\ Cov(XY) & Var(Y) \end{bmatrix} \tag{2}$$

In Eq. (2), X represents the x coordinates of all pixels in the image object, Y represents the y coordinates of all pixels in the image object, Var(X) and Var(Y) represent the variance in X and Y, respectively, and Cov(XY) represents the covariance. The aspect ratio, shape index, density, compactness, and asymmetry are adopted in our experiment. Notably, the aspect ratio refers to the proportion of the height and width of the segmentation object. The shape index describes the smoothness of the image object border. The smoother the boundary is, the lower the shape index. Density describes the distribution in the pixel space of an image object. The "densest" shape is a square; the more objects that resemble filaments, the less dense they are. Compactness describes the compactness of image objects; it is similar to the boundary index but based on area. The tighter the image object is, the smaller the border. Asymmetry describes the relative length of an image object compared with a





normal polygon. An ellipse is similar to a given image object, and it can be expressed by the ratio of the length of its small axis to that of its large axis. The eigenvalues increase with this asymmetry.

## 4.3 Attribute reduction

There are many characteristics of seismic damage in multi-source remote sensing images; thus, we should consider as many features as possible. However, the correlation between some factor characteristics and performances and the causal relationship of earthquakes is not large, and including these factors results in redundant data and makes the task onerous and meaningless. Therefore, selecting as few factors as possible is necessary while also determining the factors that best characterize the buildings.

Rough set theory is used to reduce the features in our experiment. Rough set theory was put forward in the early 1980s by Pawlak, who is an expert at the University of Warsaw, Poland, and is mainly used to study the learning, expression, and induction of incomplete data and imprecise knowledge (Pawlak, 1991; Ramanna et al., 2002; Chen and Qian, 2006; Zheng and Jin, 2010). A rough set is used to delete irrelevant or unimportant redundant data, and then knowledge discovery and mining is carried out on the premise of maintaining the ability to generalize the knowledge base. The most important feature of a rough set is that it can provide core knowledge of the data and reduce the complexity of the spatial cognition of a complex system.

## 4.4 Model element determination

An appropriate model element is the basis of the spatial assessment of earthquake damage to buildings and has an important influence on the evaluation results. In our experiment, multi-resolution segmentation is used to obtain the model element. Multiscale segmentation is a bottom-up approach and is achieved by merging adjacent pixels or small segmented objects under the premise that the average heterogeneity between the objects is the smallest and the homogeneity of the inner pixels is the largest (Definiens Image Company, 2004). Multiscale segmentation has been widely used in different types of image segmentation.

To obtain image objects of the same size, we use the multi-data joint segmentation strategy. The image segmentation scale is set to 60. Due to interference factors, there are some differences between the object range and the range of the building, and a method of comparing the building area with the image object is proposed in this paper. When 2/3 of the area of an image object is located in the position of a building, the object is considered a building; when the area is less than 2/3, the image object is excluded.

## 4.5 LRM

Logistic regression analysis is a statistical analysis method for the two categorical dependent variables (the dependent variable y takes only two values: 1 and 0 or yes and no) (Hosmer and Lemeshow, 1989; Wang and Guo, 2001). We assume that P represents the probability of destroyed buildings and Q represents the probability of intact buildings. $x_1$, $x_2$......, $x_n$



represents the N features of multi-source remote sensing images. The probability formula of destroyed buildings and intact buildings using logistic regression is as follows:

$$P = \frac{e^{a+b_1 x_1 + b_2 x_2 + \cdots + b_n x_n}}{1 + e^{a+b_1 x_1 + b_2 x_2 + \cdots + b_n x_n}} \tag{3}$$

$$Q = \frac{1}{1 + e^{a+b_1 x_1 + b_2 x_2 + \cdots + b_n x_n}} \tag{4}$$

The relationship between the probability of occurrence of an event and the influencing factors is obtained via comparison between Eq. (3) and Eq. (4). Eq. (3) is divided by Eq. (4) and then taken as the natural logarithm:

$$\ln\left(\frac{P}{Q}\right) = f(x) = a + b_1 x_1 + b_2 x_2 + \cdots + b_n x_n \tag{5}$$

In Eq. (5), A is a constant, and $b_1, b_2, \cdots b_n$ are the logistic regression coefficients. Therefore, the logistic regression analysis method can be used to establish the quantitative evaluation model. The probability P is used as the earthquake damage

assessment index (EDAI) to evaluate the ability of a feature factor $x_1, x_2, \cdots x_n$ to represent whether buildings are destroyed. Then, the spatial distribution of seismic damage in buildings is evaluated. The Statistical Package of Social Sciences (SPSS) is used to determine the relationship between calculated factors and buildings with different damage degrees.

## 5. Results and discussion

### 5.1 Features statistics and analysis

To construct the regression analysis model, 18 feature factors are selected. The feature factors are calculated using eCognition software. Prior to the calculation, the spatial resolution of the three types of remote sensing images in the study area is resampled to 1 m. The image of the study area contains 548 columns and 889 rows, and the number of pixels is 487172.

The spectral features are mainly derived from the optical imagery, including maximum, minimum, mean, brightness, and

standard deviation. Each eigenvalue is divided into 6 categories to calculate the percentage of destroyed building pixels in each category. Figure 6 illustrates the distribution of destroyed building pixels in each of the features. In the maximum value distribution, more than 15% of the destroyed buildings are mainly distributed between 91 and 150. In the minimum value distribution, the destroyed buildings are evenly distributed. In the mean value distribution, 12% of the buildings are mainly distributed between 110 and 140; in the brightness value distribution, the destroyed buildings are mainly distributed in the

range from 80–160. The value of the brightness is low, which is consistent with the presence of destroyed buildings in the optical imagery.



The texture features are mainly derived from the SAR imagery. Considering factors such as the image resolution and the distribution of seismic damage to buildings, the features of this experiment include mean, covariance, homogeneity, dissimilarity, entropy, ASM, correlation, and contrast.

Figure 7 illustrates the density distribution of destroyed buildings with different textures. When the mean value of the two-order moment is 0.1–0.2, the density of the destroyed building is larges, reaching 15%. The density distribution of seismic damage to buildings can reach 12% when the covariance is located in the range from 6000–8000. In the homogeneous image, the destroyed buildings are mainly distributed in the range from 35–50. In the dissimilarity image, all buildings are distributed in the range from 0.01–0.15, and the highest proportion is located in the range from 0.05–0.1; the proportion is approximately 11%. In the ASM image, destroyed buildings are mainly distributed in the range from 100–140. In the contrast image, destroyed buildings are mainly distributed in the range from 20–30. In the correlation image, the density distribution of destroyed buildings increases with increasing correlation and reaches the highest value at 0.3–0.36. Subsequently, the density decreases gradually.

The geometric features are mainly derived from the DSM imagery. The features include elevation, length-width ratio, shape index, density, asymmetry, and rectangularity. Figure 8 illustrates the density distribution of buildings with different geometric features. Figure 8(a) shows that the density distribution of seismic damage to buildings decreases with an increasing length-width ratio. When the aspect ratio is 1.1–1.5, which occurs in approximately 8% of the buildings, the density is largest. Figure 8(b) illustrates that the destroyed building density is largest when the shape index is in the range of 5.0–6, which is approximately 10.8%. Figure 8(c) illustrates that the density of seismic damage to buildings decreases with increasing density. In the range from 0.39–1.2, the density increases to 10%. Figure 8(d) illustrates that when the asymmetry is small (0–0.015), the density distribution of the buildings is the largest. Figure 8(e) illustrates that the density distribution of seismic damage to buildings is the largest when the rectangularity value is in the range from 0.1–0.4. Figure 8(f) illustrates that the buildings are mainly distributed at low elevations. At a height range from 590–600, the density of destroyed buildings reaches 12.3%, which is consistent with the change in the height after the building is destroyed.

After segmentation, the study area is divided into 1032 object units. The corresponding attribute of the 18 feature factors and the decision attribute corresponding to the building damage (1 represents a destroyed building, and 0 represents an intact building) form a two-dimensional table. In the table, each row describes an object and corresponds to the features of the corresponding object. The two-dimensional table contains 1032 rows and 19 columns. The initial decision table is formed by the random sampling of 10% of the table. The reduced set of spatial variations in earthquake-damaged buildings is calculated using a random sample. Figure 9 illustrates the number of different factors in the reduction. The more times the factor appears, the greater the correlation between the factor attribute and the observed earthquake damage. We remove the 11 attributes that appear least often in the reduction set and obtain a feature set consisting of 7 feature factors in our experiment. Attribute reduction is calculated using RSES software.





## 5.2 Identification model construction

Before the main statistical analyses, the data must be normalized to eliminate the effects of different data dimensions. In this study, we use a standardized processing approach to convert the value of the characteristic factor to a range from −1 to 1. The sample size is also an important factor in model construction. Generally, using similar proportions of 1 ("destroyed

building") and 0 ("intact building") cells is recommended. Hence, we took 56 random samples consisting of 26 destroyed building cells and 30 intact building cells (Figure 10).

To avoid the multicollinearity problem between explanatory variables, the selected stepwise method is based on the Wald statistic. Entry testing is performed based on the significance of the score statistics. The logistic regression mathematical equations are formulated using all factors. The following equation is obtained:

$$f(x) = 1.093 \cdot BR + 0.419 \cdot CON + 0.027 \cdot ASM + 0.076 \cdot ELE + 0.183 \cdot ASY + 0.960 \cdot HOM + 0.868 \cdot SPI - 5.291$$

where $f(x)$ represents the degree of earthquake damage for each object. The statistical significance of each coefficient in the model is listed in Table 1.

## 5.3 Results and verification

To verify the validity of the method proposed in this paper, we analyse different data combinations in accordance with the

technological process. The compound modes are as follows: (1) optical image, (2) SAR image, (3) DSM image, (4) optical image + SAR image, (5) optical image + DSM image, and (6) SAR image + DSM image.

The overall statistics of these models are shown in Table 2. The Hosmer-Lemeshow chi-square index, which is an important index for evaluating the goodness of fit of the model, is obtained by calculating the difference between the observed and predicted values of the dependent variable (Menard, 1995; Clark and Hosking, 1986). The smaller the value is, the better the

model fit. The greater the value of the −2 log-likelihood is, the better the correlation between the selected feature factors and the assessment events. The greater the value is, the higher the Cox-Snell R-square value, and the better the performance of the model.

Receiver operating characteristic (ROC) analysis is used to summarize the performance of the LRM. ROC, which is a comprehensive index reflecting sensitivity and specificity, is a method used to reveal the relationship between sensitivity and

the proportion of false negatives (Zweig and Campbell, 1993). The area under the ROC curve indicates the accuracy of the model. Theoretically, the measure has a value from 0.5 to 1. When the value is 1, the evaluation precision is highest, while a value of 0.5 indicates that the evaluation is worthless (Lulseged and Hiromit, 2005). The area under the ROC curve that corresponds to our study (0.827) is shown in Figure 11. The result of the analysis that considers the multi-source remote sensing image is better than the results of other combinations. In addition, the evaluation of the combined SAR image and

optical image is the best. The single DSM image evaluation results in the lowest.

 (1) Classification accuracy comparison



Twenty-nine intact buildings and six destroyed buildings are chosen to test the accuracy of the model. The evaluation results of different features are compared with the results of the field survey, and then, the model accuracy is analysed. Table 3 presents the classification results of the seismic damage assessment for different data sources. Half of the collapsed buildings are mistaken for intact buildings in when using the DSM image. For the combinations of the optical image with the SAR

image and the optical image with the DSM image, the false identification of destroyed buildings is reduced, and the overall accuracy is 85.71% and 88.57%, respectively. For the combination of the SAR image with the optical image, the destroyed buildings are easily identified; the overall classification accuracy is 91.43%, which illustrates that the combination of texture features and geometric features can better identify damaged buildings. Combining the three different sources of data, only one destroyed building is misclassified as an intact building. In addition, one intact building is misclassified as a destroyed

building (Figure 12); thus, the overall classification accuracy is 94.2%, which illustrates that the combination of multi-source data can effectively improve the classification accuracy of seismic damage information.

(2) Quantitative analysis of evaluation precision

To analyse the potential effect of the proposed model, the calculated earthquake damage building maps are compared with the field investigation results. Accuracy curves are used to show the performance (Aleotti and Chowdhury, 1999; Chung and

Fabbri, 1999). To obtain the accuracy curves, the earthquake damage assessment is sorted in descending order. The number of building cells within the period (1–100% with accumulated 1% intervals) for each class is counted. The relationship between the EDAI rank and the cumulative percentage of destroyed buildings is shown in Figure 13. The EDAI rank (x-axis) ranges from a high to low level of seismic damage (i.e., from destroyed to intact). Therefore, a low EDAI rank indicates higher earthquake damage to a building. For example, a 20% EDAI rank in the study area accounts for 65% cumulative

damage. In addition, a 30% EDAI rank in the study area accounts for 79% cumulative building damage.

## 5.4 Discussion

In the experiment, 18 characteristics of spectral features, texture features, and shape features are selected for analysis, and their ability to characterize earthquake-related building is calculated based on rough set theory. Although three kinds of features can be extracted by optical imagery or SAR imagery, the amount of information contained by different datasets

varies. Compared with optical imagery, SAR imagery has reduced spectral performance, although the texture information is relatively abundant, and the fusion of multi-source and multi-feature data can compensate for the deficiencies in a single data source.

The logistic method proposed in our experiment is mainly used to construct a prediction model. The method is based on the influence of multiple factors on a certain event, and the probability of model prediction is established. In this paper, the

model is applied for the extraction of intact buildings and destroyed buildings after an earthquake. The model establishes a quantitative relationship between the EDAI and different features. During model construction, the division of the model calculation unit is the basis of evaluation. The usual method of unit computation is the grid element partition of the raster



data structure. In this study, the objects segmented by multiscale segmentation are used as evaluation units, which can better describe the characteristics of buildings and express them better by means of evaluation factors.

The validity of the proposed method is verified by using the earthquake site of old Beichuan County as a research area, and the recognition accuracy of the three kinds of data features reaches 94.2%. In this experiment, data from the Beichuan

earthquake site are used; additionally, the site was renovated, and the buildings were cleaned up after the earthquake. The buildings are divided into standing buildings and destroyed buildings, which are limited by the distribution of the earthquake site buildings in old Beichuan County. The method proposed in this paper is based on the sample, and the EDAI can classify seismic damage to buildings in detail if a more detailed classification system of seismic damage can be obtained.

## 6. Conclusions

In this paper, a new method for the quantitative evaluation of earthquake damage to buildings based on multi-source remote sensing data is proposed. This method can be used to evaluate earthquake-damaged buildings by constructing a regression model. The effectiveness of the proposed approach is confirmed using multi-source remote sensing images of old Beichuan County. With this method, intact buildings and destroyed buildings can be effectively distinguished. The extraction accuracy is 94.2%, and the area under the ROC curve of the model is 0.827. The method is tested using optical images, SAR images,

DSM images, and combined images to demonstrate the effectiveness of the proposed method. The results also show that the fusion of different source data can improve the classification accuracy of earthquake-damaged buildings. Model accuracy verification shows that the model has high reliability and accuracy and can be used for seismic damage assessment and analysis of buildings.

**Acknowledgements:** The authors wish to thank the anonymous reviewers for their valuable comments. This work was

20 supported by the research grant from the Institute of Crustal Dynamics, China Earthquake Administration (Grant Nos. ZDJ2018-14 and ZDJ-2017-29), the National Natural Science Foundation of China (Grant Nos. 41874059 and 41602223), and the Civil Aerospace Project (Grant No. D010102).

**Author contribution:** Jingfa Zhang and Qiang Li conceived and designed the experiments; Qiang Li performed the experiments; Jingfa Zhang and Hongbo Jiang analysed the data; Qiang Li wrote the paper.

**Disclosure statement:** The authors declare that there are no conflicts of interest regarding the publication of this paper.

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



**Figure 1. Study area. (a) Geographical location of the study area and the optical image acquired in 2013 using a UAV; (b) the building distribution acquired by the high-resolution optical image interpretation and field investigation.**





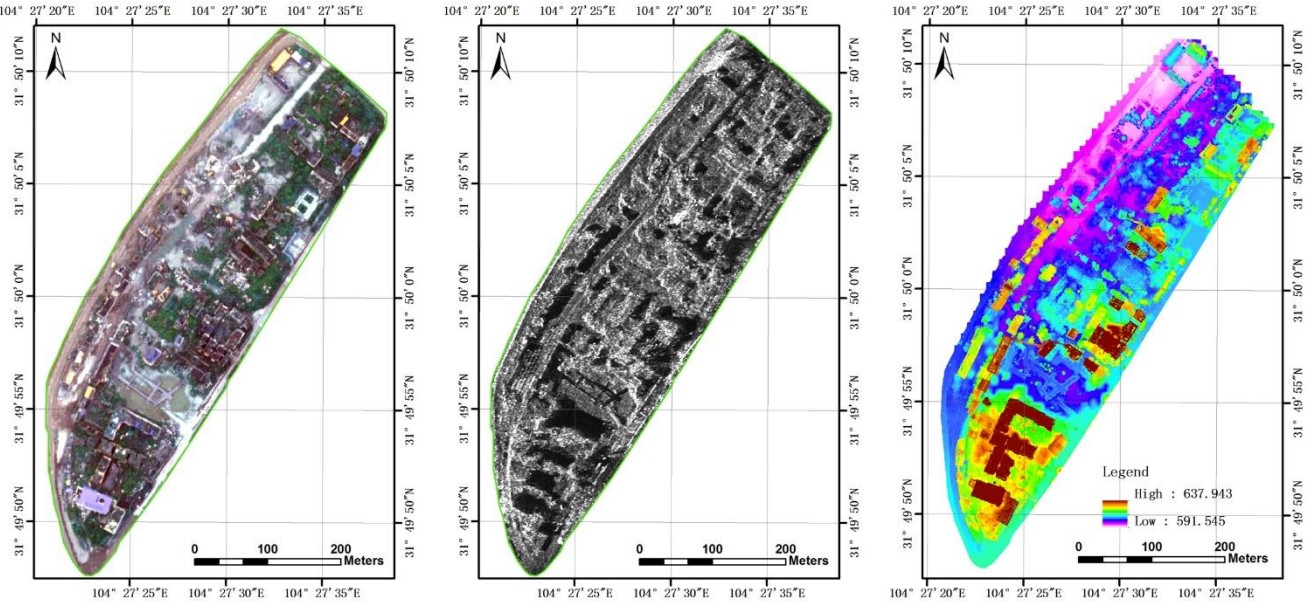

**Figure 2. Multi-source remote sensing image of the study area. (a) Airborne multispectral remote sensing image acquired in 2013; (b) TerraSAR-X ascending image (the azimuth direction is from bottom to top, and the range direction is from left to right); and (c) DSM image generated with LiDAR point cloud data interpolation.**

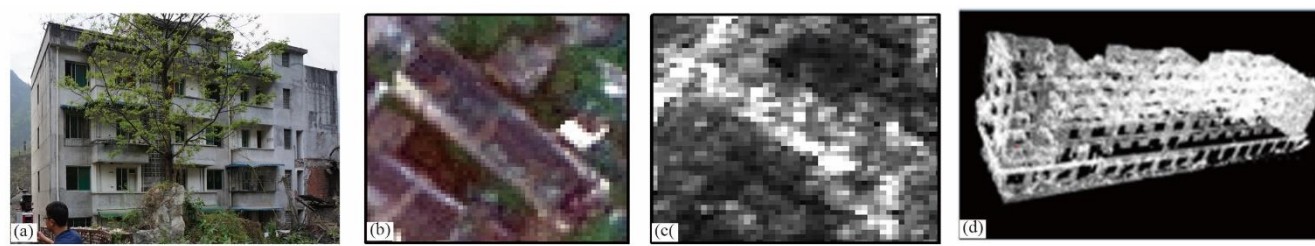

**Figure 3. Intact building after the earthquake: (a) field investigation photograph; (b) optical image acquired in 2013; (c) TerraSAR-X ascending image (the azimuth direction is from bottom to top, and the range direction is from left to right); and (d) LiDAR image acquired in 2013.**

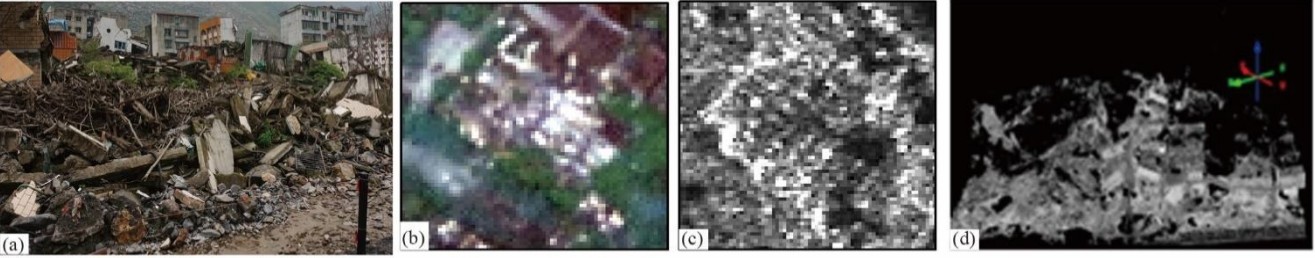



**Figure 4. Destroyed building after the earthquake: (a) field investigation photograph; (b) optical image acquired in 2013; (c) TerraSAR-X ascending image (the azimuth direction is from bottom to top, and the range direction is from left to right); and (d) LiDAR image acquired in 2013.**

**Figure 5. Technical flow of the proposed methodology.**





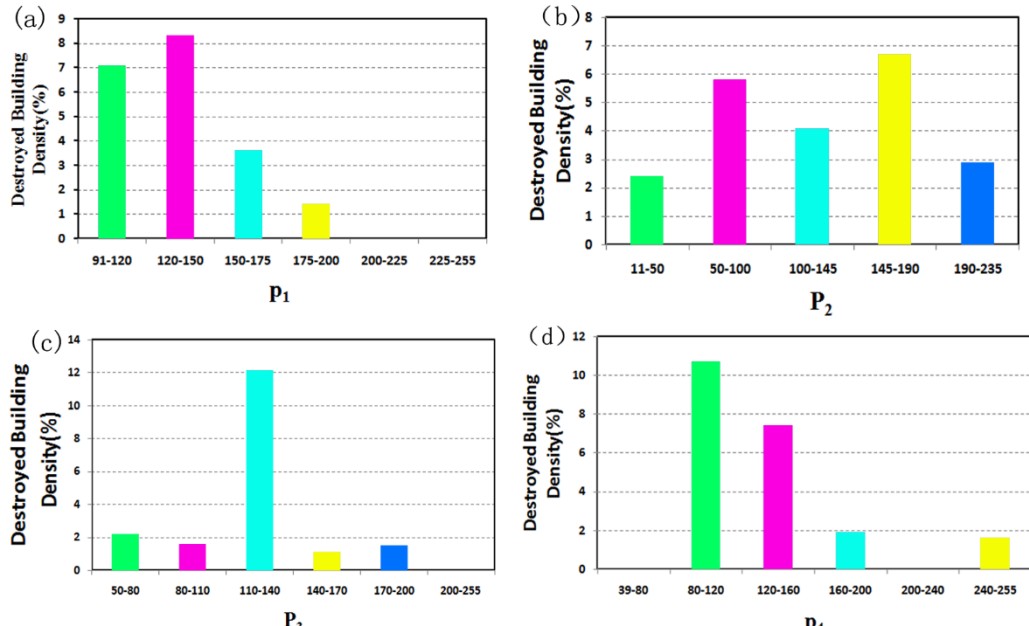

**Figure 6. Density distribution of destroyed buildings with different spectral features. (a) MAX; (b) MIN; (c) mean; and (d) brightness.**



**Figure 7. Density distribution of destroyed buildings with different texture features: (a) ME; (b) VA; (c) HOM; (d) DI; (e) ENT; (f) ASM; (g) CON; and (h) COR.**



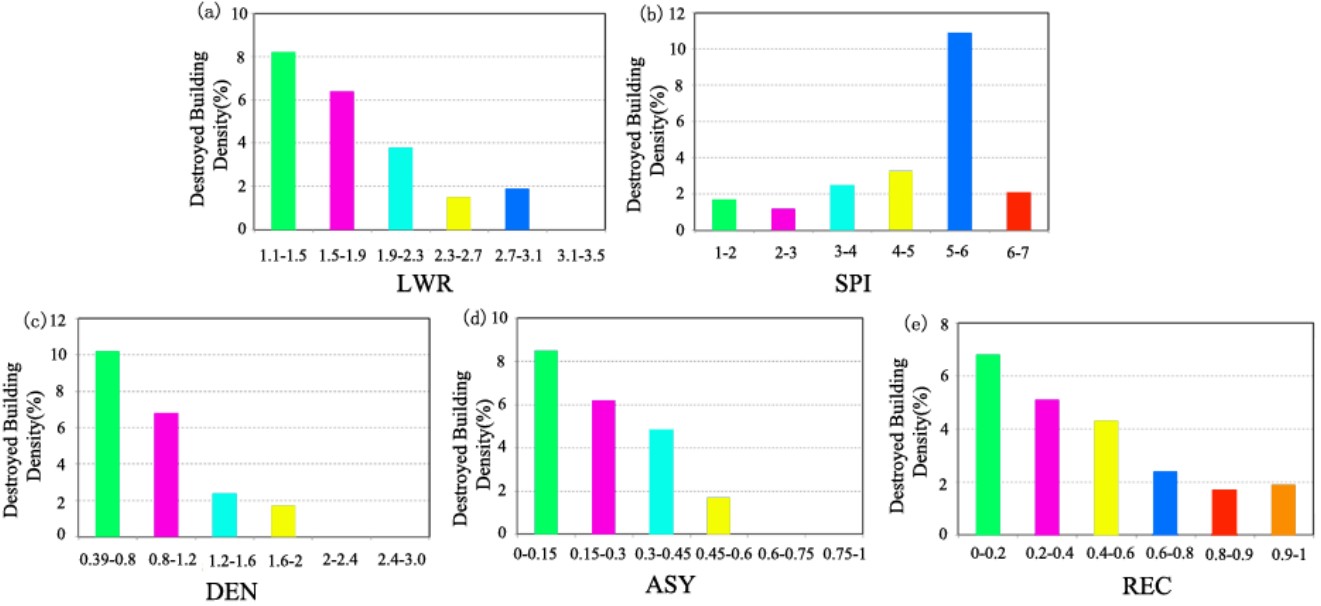

**Figure 8. Density distribution of destroyed buildings with different geometry features: (a) LWR; (b) SPI; (c) DEN; (d) ASY; (e) REC; and (f) ELE.**

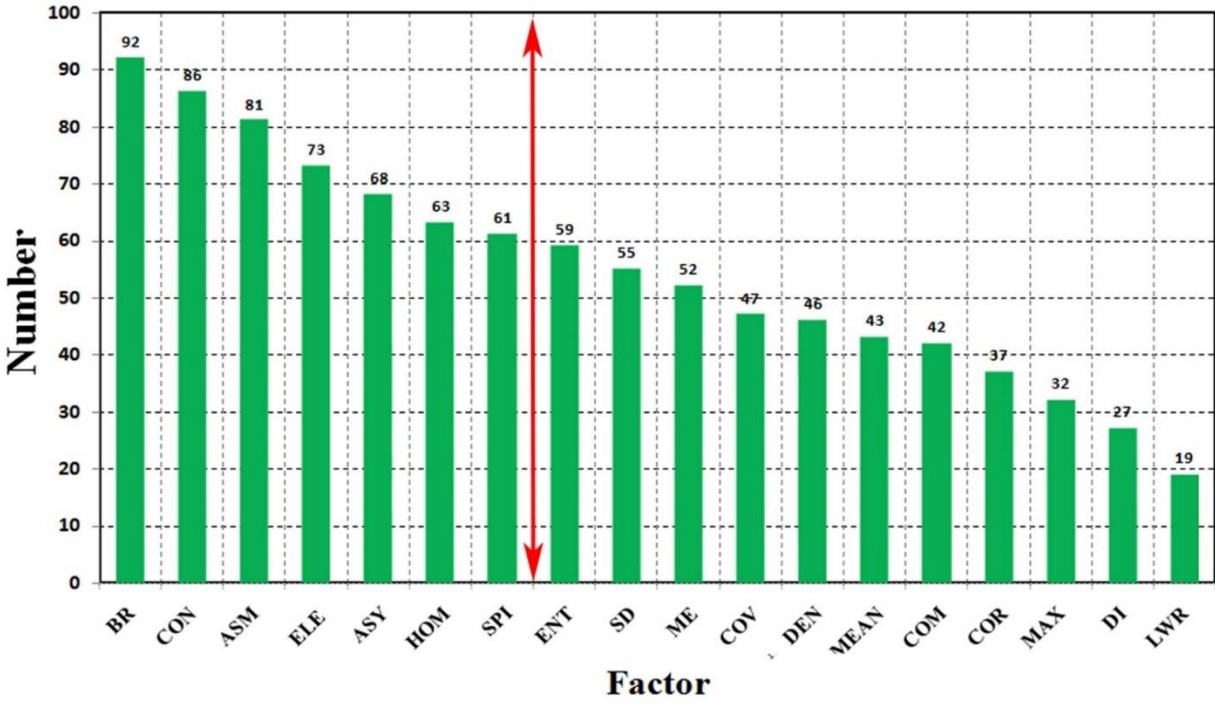

**Figure 9. Attribute reduction set. The number represents the frequency of the feature, and the red line represents the feature set of the segment selection.**



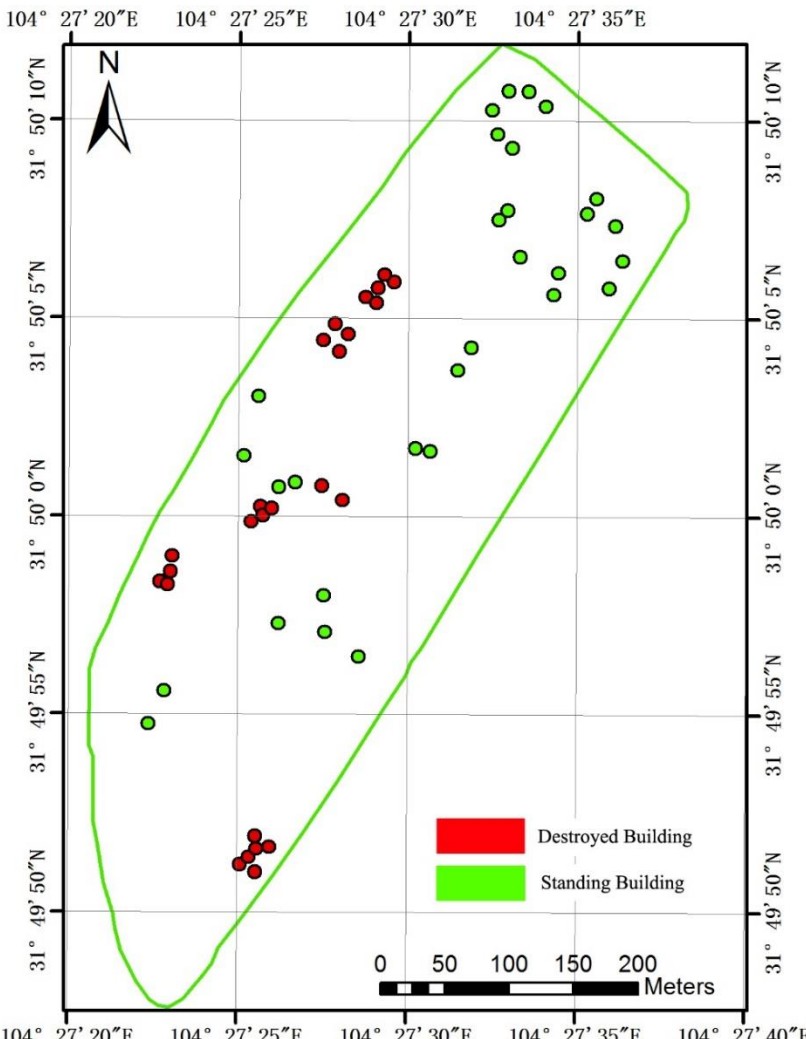

**Figure 10. Distribution of samples in the logistic regression.**





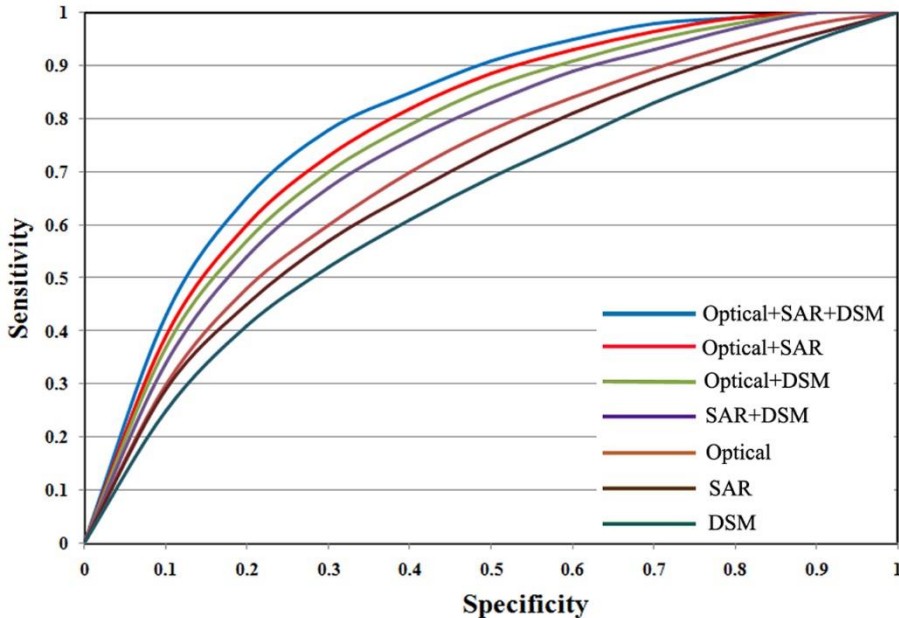

**Figure 11. Receiver operating characteristic (ROC) curve.**





**Figure 12. Experimental results and field investigation results of earthquake-damaged buildings: (a) earthquake-damaged building map based on the logistic regression; and (b) earthquake-damaged building map based on high-resolution optical image interpretation and field investigation.**





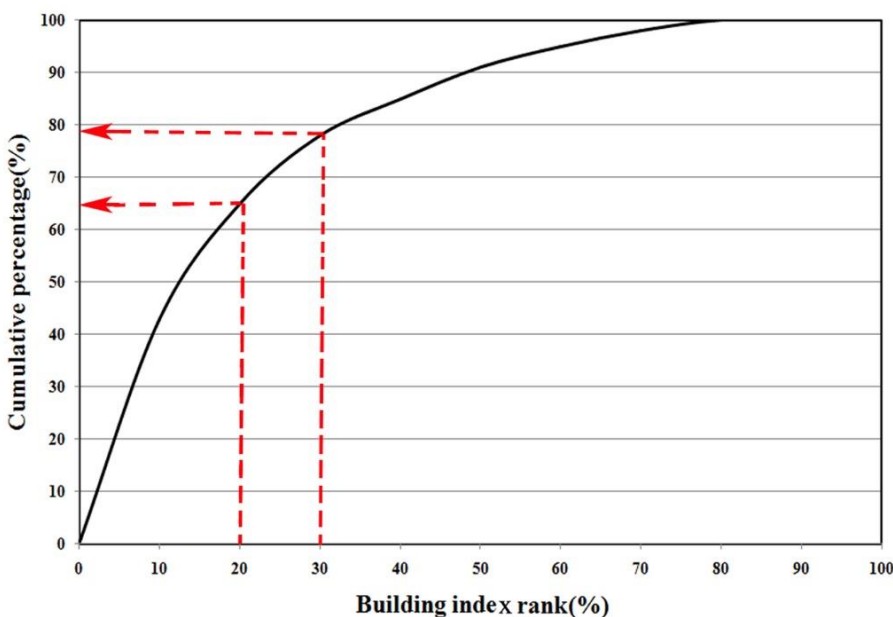

**Figure 13. Cumulative frequency diagram showing the cumulative building index rank relative to the cumulative percentage of destroyed buildings.**

5    **Table 1. Logistic regression results and coefficient values used for this study**

| Variable | B | S.E. | Wald | Exp(B) | Sig. |
|---|---|---|---|---|---|
| BR | 1.093 | 0.135 | 64.49 | 2.983 | 0.000 |
| CON | 0.419 | 0.107 | 14.32 | 1.520 | 0.000 |
| ASM | 0.027 | 0.120 | 0.050 | 1.027 | 0.000 |
| ELE | 0.076 | 0.031 | 5.98 | 1.078 | 0.000 |
| ASY | 0.183 | 0.248 | 0.494 | 1.200 | 0.015 |
| HOM | 0.960 | 0.231 | 17.02 | 2.611 | 0.000 |
| SPI | 0.868 | 0.104 | 68.82 | 2.382 | 0.030 |





**Table 2. Overall statistics of the LRM**

| Mode | -2 Log-likelihood | Cox-Snell R-Squared | Hosmer-Lemeshow Chi-square |
|---|---|---|---|
| 1 | 6491.077 | 0.509 | 15.001 |
| 2 | 6328.167 | 0.496 | 18.926 |
| 3 | 6309.297 | 0.484 | 15.324 |
| 4 | 6970.789 | 0.571 | 13.151 |
| 5 | 6869.043 | 0.535 | 14.746 |
| 6 | 6682.480 | 0.526 | 14.003 |
| 7 | 7070.968 | 0.592 | 13.085 |

5   **Table 3. Assessment results with different combination modes (Pa=producer's accuracy; Ua=user's accuracy)**

| Method | Destroyed Building | | Standing Building | | Overall |
|---|---|---|---|---|---|
| | Pa | Ua | Pa | Ua | |
| DSM | 42.86 | 50 | 89.29 | 86.21 | 80 |
| SAR | 46.15 | 100 | 100 | 75.86 | 80 |
| Optical | 57.14 | 66.67 | 92.86 | 89.66 | 85.71 |
| SAR+DSM | 57.14 | 66.67 | 92.86 | 89.66 | 85.71 |
| Optical+DSM | 62.5 | 83.33 | 96.30 | 89.66 | 88.57 |
| Optical+SAR | 66.67 | 100 | 100 | 89.66 | 91.43 |
| Optical+SAR+DSM | 83.33 | 83.33 | 96.55 | 96.55 | 94.2 |