# Peer review of "Incorporating multi-source remote sensing in the detection of earthquake-damaged buildings based on logistic regression modelling"

_Natural Hazards and Earth System Sciences, 2019_

## Referee Comment (RC1) · Anonymous Referee #1 · 22 Oct 2019

The work proposes to detect earthquake-damaged buildings using multi-source remote sensing data. The proposed work integrates the use of data acquired using multispectracal sensors, SAR and LiDAR.

The written content of the paper is quite comprehensible even if the text should be revised by a mother tongue. The paper is technically fair and the topic is relevant for the journal.

However, the following points need to be clarified and eventually revised:

Abstract: page 1 line 10 - different "modes" should be changed with different "types" as it is more appropriate page 1 line 22 - "which is be regarded" change to "which is

regarded" page 1 line 28 - by features do you mean data sources?

2. Study case and datasets page 4 row 21-22 - what type of sensor did you used? what is the date and time of the acquisition? page 4 row 24 - the authors mention to use a 0.5 m image but at page 7 row 1 the resolution changes to 0.2. Please clarify page 4 row 27 - please provide date and time of the acquisition

3.Seismic characteristics of multi-source remote sensing images page 5 row 19 - "the square nature of the structure", is not clear, do you mean the "ground projection"?

4.1 Data processing page 7 row 5 - please cite in an appropriate way Google maps (here you can find a hint:http://writeanswers.royalroads.ca/faq/199225)

4.1 Data processing page 7 row 21 - please explain better what do you mean by geometric feature

4.4 Model element determination page 8 row 26-27 - is this an empiric assumption?

5.1 Features statistics and analysis page 9 row 17 - all images are resampled to 1 m. What is the method used?(nearest neighbour)

5.3 Results and verification page 11 row 27-30 - the authors state that "the moultisource remote sensing image is better than the results of other combinations" but soon after you say that the "combined SAR image and optical image is the best". It's not clear which method is more accurate.

5.3 Results and verification page 12 row 10 - please pin point the building in the image or write it's geographical coordinates

Figure 2 please mention the date and time stamp of each image (hear or in the text)

Figure 3 please mention the geographical coordinates of the building

Figure 4 please mention the geographical coordinates of the building

Figure 8 (f) ELE parameter is missing

Figure 9 parameter BR and ELE are never mentioned in the text or in the other figures

[Figure]

---

## Author Comment (AC1) · 25 Nov 2019

1.Abstract: page 1 line 10 - different "modes" should be changed with different "types" as it is more appropriate. page 1 line 22 - "which is be regarded" change to "which is regarded". page 1 line 28 - by features do you mean data sources? Reply: Thanks for the reviewer's suggestion. We have changed it in the revised paper. Yes, we do mean data sources. 2. Study case and datasets page 4 row 21-22 - what type of sensor did you used? what is the date and time of the acquisition? page 4 row 24 - the authors mention to use a 0.5 m image but at page 7 row 1 the resolution changes to 0.2. Please clarify page 4 row 27 - please provide date and time of the acquisition

Reply: Sorry. We have added the types of sensor and the time of the acquisition in the revised paper. Sorry, we have changed the resolution of the image to 0.5m. 3.Seismic characteristics of multi-source remote sensing images page 5 row 19 - "the square nature of the structure", is not clear, do you mean the "ground projection"? Reply: Sorry, "the square nature of the structure" in the paper means the shape characteristics of building structure, we changed in the revised paper. 4. 4.1 Data processing page 7 row 5 - please cite in an appropriate way Google maps (here you can find a hint:http://writeanswers.royalroads.ca/faq/199225) Reply: Thanks for the reviewer's suggestion. We have changed it in the revised paper. 5. 4.1 Data processing page 7 row 21 - please explain better what do you mean by geometric feature Reply: Geometric features are the most direct physical attribute of targets. The unit we analyze shape features is the object. It usually contains the area, perimeter and other information of the targets. Aspect ratio, shape index, density, compactness, and asymmetry are adopted in our experiment. The shape index describes the smoothness of the image object border. Density describes the distribution in the pixel 20 space of an image object. Compactness describes the compactness of image objects; it is similar to the boundary index but based on area. Asymmetry describes the relative length of an image object compared with a normal polygon. 6. 4.4 Model element determination page 8 row 26-27 - is this an empiric assumption? Reply: YesïïjŇthe method of model element determination is empiric assumption. After many experiments, we analyze the segmented objects to determine the optimal results. 7. 5.1 Features statistics and analysis page 9 row 17 - all images are resampled to 1 m. What is the method used?(nearest neighbour) Reply: Yes, the resample method used in this paper is nearest neighbor. 8. 5.3 Results and verification page 11 row 27-30 - the authors state that "the moultisource remote sensing image is better than the results of other combinations" but soon after you say that the "combined SAR image and optical image is the best". It's not clear which method is more accurate. Reply: Sorry about it. The result of the analysis that considers the multi-source remote sensing image is best among the results of other combinations. The result of combining SAR image and optical image

comes second. 9. 5.3 Results and verification page 12 row 10 - please pin point the building in the image or write it's geographical coordinates Reply: Thank you. We have added the geographical coordinates in the revised paper. 10. Figure 2 please mention the date and time stamp of each image (hear or in the text) Reply: Thank you. We have added the date and time stamp of each image in the revised paper. 10. Figure 3 please mention the geographical coordinates of the building Reply: Thank you. We have added the date and time stamp of each image in the revised paper. Intact building after the earthquakeïijĹcentered at approximately 104.46°E and 31.83°NïijĹ 11. Figure 4 please mention the geographical coordinates of the building Reply: Thank you. We have added the date and time stamp of each image in the revised paper. Destroyed building after the earthquakeïijĹcentered at approximately 104.45°E and 31.84°NïijĹ. 12. Figure 8 (f) ELE parameter is missing Reply: Sorry about it. We have added it in the revised paper. Please refer to the attachment for the revised contents 13. Figure 9 parameter BR and ELE are never mentioned in the text or in the other figures. Reply: Sorry about it. BR means the brightness of spectral features in optical images. The specific calculation method is not introduced in this paper, but is given in the form of references. ELE means the elevation. The expression in the paper is as follows "In addition to elevation information, the main feature of the DSM image is the geometric feature." The above two abbreviations are marked in the revised paper.

The full text will be revised by a mother tongue.
* * *
**Fig. 1.** Density distribution of destroyed buildings with different spectral features. (a) MAX; (b) MIN; (c) mean; and (d) brightness.

[Figure]

**Fig. 2.** Density distribution of destroyed buildings with different geometry features: (a) LWR; (b) SPI; (c) DEN; (d) ASY; (e) REC; and (f) ELE

Table1.Detail image information

| Number | Data type | Type of Sensor | Date of Acquisition |
|---|---|---|---|
| 1 | Optical multispectral image | UAV | 4 July 2013 |
| 2 | LiDAR | RIEGL VZ2000 | 20 July, 2014 |
| 3 | SAR data | TERRASAR | 4 December 2014 |

**Fig. 3.** Table of detail image information

---

## Short Comment (SC1) · 7 May 2020

In this paper, a method of building damage extraction based on multi-source remote sensing data is proposed and applied to Beichuan area, which is a good innovation. The paper is technically fair and the topic is relevant for the journal. However, there are also some issues that need to be revised: 1.In the abstract part, the author introduces the application of single source data in the extraction of seismic damage information. As far as I know, many scholars have also begun to study the methods of extracting information from images by fusion of two or more types of data. It is suggested that the author supplement and supplement relevant references. 2.In the part of datasets

introduction, it is suggested that the author take the form of table, which can be more intuitive. 3ïïjŐThe author analyzes the characteristics of different buildings and adopts the object analysis method, so how does the author obtain the image object and what is the segmentation method adopted? It is suggested that the author should supplement it in the original text 4.The logical regression method used in this paper is a statistical analysis method, so samples are needed. Please elaborate the principles and methods of sample selection. 5.What is the method of feature selection? It is suggested to add a detailed description. 6.What does the table mean in"The initial decision table is formed by the random sampling of a 10% table." 7.In Figure 8. The ELE is missing. 8ïïjŐIn Fig. 7 and Fig. 8, the abbreviations are used, while in Fig. 6, P1-P4 is used, so it is recommended to be unified. 9.There are some references with different formats

---

## Author Comment (AC2) · 13 May 2020

In this paper, a method of building damage extraction based on multi-source remote sensing data is proposed and applied to Beichuan area, which is a good innovation. The paper is technically fair and the topic is relevant for the journal. However, there are also some issues that need to be revised: 1.In the abstract part, the author introduces the application of single source data in the extraction of seismic damage information. As far as I know, many scholars have also begun to study the methods of extracting information from images by fusion of two or more types of data. It is suggested that the author supplement and supplement relevant references. ReplyïijŽThanks for

the reviewer's suggestion. We will add the relevant references in the revised paper. The following references were added: Yinyi Lin,Hongsheng Zhang,Hui Lin,Paolo Ettore Gamba,Xiaoping Liu. Incorporating synthetic aperture radar and optical images to investigate the annual dynamics of anthropogenic impervious surface at large scale. Remote Sensing of Environment,2020,242. Fang Chen,Zhigang Yuan,Yongfeng Huang. Multi-source data fusion for aspect-level sentiment classification. Knowledge-Based Systems,2020,187. Dino Ienco,Roberto Interdonato,Raffaele Gaetano,Dinh Ho Tong Minh. Combining Sentinel-1 and Sentinel-2 Satellite Image Time Series for land cover mapping via a multi-source deep learning architecture. ISPRS Journal of Photogrammetry and Remote Sensing,2019,158. Shiran Song, Jianhua Liu, Yuan Liu, et al. Intelligent Object Recognition of Urban Water Bodies Based on Deep Learning for Multi-Source and Multi-Temporal High Spatial Resolution Remote Sensing Imagery. 2020, 20(2) AI Jinquan. Long-term evolution process and mechanisms of wetland ecosystem in the Yangtze River estuary using time-series multi-sensor remote sensing data. Acta Geodaetica et Cartographica Sinica, 2020, 49(1): 133-133. 2.In the part of datasets introduction, it is suggested that the author take the form of table, which can be more intuitive. RelplyïijŽThanks for the reviewer's suggestion. We will turn it into a table in the revised paper. 3ïijŐThe author analyzes the characteristics of different buildings and adopts the object analysis method, so how does the author obtain the image object and what is the segmentation method adopted? It is suggested that the author should supplement it in the original text Reply: Thanks for the reviewer's suggestion. The multi-scale segmentation is used to obtain the model element. Multiscale segmentation is a bottom-up approach and is achieved by merging adjacent pixels or small segmented objects under the premise that the average heterogeneity between the objects is the smallest and the homogeneity of the inner pixels is the largest. To obtain image objects of the same size, we use the multi-data joint segmentation strategy. The image segmentation scale is set to 60. 4.The logical regression method used in this paper is a statistical analysis method, so samples are needed. Please elaborate the principles and methods of sample selection. Reply: Thanks for the reviewer's suggestion. In the manuscript, the introduction of principles and methods of sample selection is as follows: Generally, using similar proportions of 1 ("destroyed building") and 0 ("intact building") cells is recommended. Hence, we took 56 random samples consisting of 26 destroyed building cells and 30 intact building cells. 5.What is the method of feature selection? It is suggested to add a detailed description. Reply: Thanks for the reviewer's suggestion. We selected the object characteristics based on some previous published papers and related literature. We will add them in the revised paper. 6.What does the table mean in"The initial decision table is formed by the random sampling of a 10% table." The corresponding attribute of the 18 feature factors and the decision attribute corresponding to the damage of the building (1 represents a destroyed building, and 0 represents an intact building) form a two-dimensional table. In the table, each row describes an object and corresponds to the features of the corresponding object; that is, the two-dimensional table contains 1032 rows and 19 columns. The initial decision table contains 10% of the sample information. 7.In Figure 8. The ELE is missing. Reply: Thanks for the reviewer's suggestion. We have added it in the revised paper. 8.ïijŐIn Fig. 7 and Fig. 8, the abbreviations are used, while in Fig. 6, P1-P4 is used, so it is recommended to be unified. Reply: Sorry about it. We will change them in the revised paper. 9.There are some references with different formats Reply: Sorry about it. We will modify it according to the requirements of the paper template.

---

## Referee Comment (RC2) · Anonymous Referee #2 · 24 Aug 2020

This is a review of "Incorporating multi-source remote sensing in the detection of earthquake-damaged buildings based on logistic regression modelling"

Overall, this is a fairly straight forward examination of remote sensing images with ground truthing of earthquake damage using logistic regression. Although much of the way there, the paper needs work to bring it up to an international level of science in terms of formatting, English, structure, referencing of other authors, and convince us this goes beyond a case study. Overall, with a major revision this should be acceptable.

Comments (not in order of importance):

• ABSTRACT. The abstract is very wordy, and lacks, until we get to the last few

sentences, a quantitative description of the data, methods, analysis. At no point in the abstract does it talk about which years/how many images, how big an area is studied, but rather is a narrative description of the data. Please make this more of a summary of the manuscript, rather than narrative. • English. Throughout, this will need to be checked carefully by the copy editors, but overall, the English is understandable (but English as a second language). • Paragraph 1 (Introduction). Please add appropriate references (cite other people), rather than just narrative. • Introduction (background). I did not feel that you have appropriately reflected the literature of OTH-ERS that have done work on multi-source remote sensing for earthquake damaged buildings. I would like to suggest either in the introduction, or another section which should be named BACKGROUND (or something similar) you do a much more thorough literature review of those who have worked on examining earthquake damaged buildings based on remote sensing. Ideally, this would be a TABLE with headers that pull out information from these papers, and provides a critical review (it does not have to be at a review paper level, but enough so we have an idea of what has been done before). These headers might be "Source" (e.g., Voigt et al, 2007), Region, Earthquake, Remote Sensing Products Used, . . ., . . .., . . .., Main comments. Then, in the text of the paper, you can refer to this table, and compare and contrast. As it is, the studies you cite tend to be dated (2007, 2009, 2011, 2011, 2006, 2012, 2011, etc.) with no papers in the introduction which are since 2012. A lot has happened since then, and it does not feel that you are 'building' on others' work by acknowledging them. The overall result is a Master's thesis, and not critically done, in terms of the background. • References. Throughout, please go sentence by sentence and ensure that you have referred to the literature. If you have facts, ideas of other people, you need an in-text citation. For example, in Section 2, you do not have any references, but then state items of fact such as the Wenchuan earthquake caused a large number of casualties and damage to facilities (give a reference). Old Beichuan County resulted in relocation of the entire community (needs a reference). There are many similar sentences. You need to be VERY CLEAR where your facts and information that you cite are from. •

Section 2. Study case and Datasets. I'd like a lot more specificity about the study area and the data used. How big is the study area (Old Beichuan County)—what kind of geology is there? Is it an area heavily populated? Density? Lots of buildings? For the datasets, which years/months? How many? You are vague about the data, so a person would not be able to repeat what you did (they don't know what you used). Throughout, you need to ensure that the reader knows the (relevant background) to the study area, exactly the data you used, and then what you did with it. • Section 3. Seismic Characteristics of multi-source remote sensing images. This is fairly description rather than quantitative in its presentation of the seismic characteristics one can detect using remote sensing images. There are some good parts in here, but can the section be made slightly more organized in its structure. This is evident also in having just one reference cited for the entire section—has there really been no one else who has looked at seismic characteristics using multi-source remote sensing images? • Section 4. Methodology. In terms of structure, this borders on narrative in places and coule be slightly better organized in terms of "We did the following steps: (i) ****, (ii) ****, (iii) *****" with any appropriate references. In terms of content of the methodology, although parts of this are good, imagine someone who does not have your work, trying to now read it and replicate it. Have you put in enough details for that person to reproduce each step. So give this to one of your (student) colleagues NOT familiar with the work, and ask them if they could reproduce each step over an hour. • Section 5 and 6. I'd like to better understand the behaviour of your results and the uncertainty. So in practice, what would it mean if we were to use your algorithm in another region? Would we get 50% of the buildings correctly identified as damaged or not damaged? More? Yes, you give us ROC diagrams and tables of numbers, but what would this mean in practice in terms of uncertainty. This is for me the key part of the paper. You have data input, a methodology (your 'black box'), and then results–how good would those results be elsewhere and what might be limitations (e.g., if an image has clouds in it, resolution of the remote sensing image, type of land use)? A more nuanced discussion of these based on the literature of what others have done would make this into a more

far reaching paper. • Equations. Equation 1. Infoterra is an 'interesting' source. Do you genuinely have no other sources of reference for this key equation? What is sigma standing for—you have not told us, nor why it is raised to the 0th power (I guess this is your radar sigma naught value). What is s in the k_s. Some problems for this and the other equations with formatting. Tell us the range of values here, and why DN is in absolute values. So give us some feeling for this equation, and the data going into it. Equations 3 and 4 need to have a source of their information, and the formatting looks really odd. Equation 5 and text that follows it—please check your text carefully for typos, you state "A is a contast" but Eq. (5) has not "A" it has an "a". These are not the same. I'm not clear what X1, X2, etc., are (you state it is a feature factor). Give us an idea of some values for these, their range, what they look like. Ah, I see you do so later—but then you have to tell us you will do this later. I'd still like to better understand this variable x. • Variables. You seem to go back and forth between different font for variables, particularly x, and you do not consistently use italic. • Units. Please check all numbers have appropriate units, e.g., "at a height range of 590-600" [? m] • Equation in Section 5.2—give this a number, and put brackets in appropriate places (1.093*BR) + (0.419*CON". Remind us what the acronyms mean. • Conclusions. I'm not convinced whether this is a paper that really is a new method. You are somewhat vague on exisiting literature. I think overall it is good that you have done this methodology, just would like to see better convincing about what has been done by others. Overall, though, I think this will add incrementally to the literature. • FIGURES • For all figure captions, if you refer to 'data obtained' give us the source of the data (e.g., by authors, by ****, by *****). • Figure 6. Variables go back and forth between p and P. Figure caption needs to be more complete. The colours made no sense to me—what do these mean? Even going back to the text, I was unsure exactly what P1, P2, P3, P4 meant. The figure caption should be self standing, so a reader does not need to go back to the text, but you are vague here. • Figures 7-8. Define what you mean by ME, VA, HOM, DI, etc., in the figure caption. Why different colours. Poorly done labels in places—this is probably one of the least professional

figures you have in the paper. You are just giving lots of acronyms without proper explanation of what they are (and one has to read the entire paper in depth to understand these. • Figure 9. Same thing, acronyms? • Figure 10. Good • Figure 13. I didn't get this (based on your caption). • Table 3. Please use the same precision throughout. (e.g., 50.00 not 50).
* * *

---

## Author Comment (AC3) · 22 Sep 2020

Dear reviewer: I am very grateful to your comments for the manuscript. According with your advice, we amended the relevant part in manuscript. Some of your questions were answered below. 1. ABSTRACT. The abstract is very wordy, and lacks, until we get to the last few At no point in the abstract does it talk about which years/how many images, how big an area is studied, but rather is a narrative description of the data. Please make this more of a summary of the manuscript, rather than narrative. Reply: Thank you for your suggestion. We will reorganize and compile the abstract and supplement the relevant main contents. Please refer to the revised manuscript

for details 2. English. Throughout, this will need to be checked carefully by the copy editors, but overall, the English is understandable (but English as a second language). Reply: Thank you very much. We will employ a professional language polishing company to modify the language. 3.Paragraph 1 (Introduction). Please add appropriate references (cite other people), rather than just narrative. Reply: Thank you very much. We will supplement the relevant important references in the corresponding position. 4.Introduction (background). I did not feel that you have appropriately reflected the literature of OTHERS that have done work on multi-source remote sensing for earthquake damaged buildings. I would like to suggest either in the introduction, or another section which should be named BACKGROUND (or something similar) you do a much more thorough literature review of those who have worked on examining earthquake damaged buildings based on remote sensing. Ideally, this would be a TABLE with headers that pull out information from these papers, and provides a critical review (it does not have to be at a review paper level, but enough so we have an idea of what has been done before). These headers might be "Source" (e.g., Voigt et al, 2007), Region, Earthquake, Remote Sensing Products Used, : : :, : : :., : : :., Main comments. Then, in the text of the paper, you can refer to this table, and compare and contrast. As it is, the studies you cite tend to be dated (2007, 2009, 2011, 2011, 2006, 2012, 2011, etc.) with no papers in the introduction which are since 2012. A lot has happened since then, and it does not feel that you are 'building' on others' work by acknowledging them. The overall result is a Master's thesis, and not critically done, in terms of the background. References. Throughout, please go sentence by sentence and ensure that you have referred to the literature. If you have facts, ideas of other people, you need an in-text citation. For example, in Section 2, you do not have any references, but then state items of fact such as the Wenchuan earthquake caused a large number of casualties and damage to facilities (give a reference). Old Beichuan County resulted in relocation of the entire community (needs a reference). There are many similar sentences. You need to be VERY CLEAR where your facts and information that you cite are from. Reply: Thank you very much. We will carefully sort out

the full text and supplement relevant references in the quoted position. Please refer to the revised manuscript for details. 5.Section 2. Study case and Datasets. I'd like a lot more specificity about the study area and the data used. How big is the study area (Old Beichuan County) .what kind of geology is there? Is it an area heavily populated? Density? Lots of buildings? For the datasets, which years/months? How many? You are vague about the data, so person would not be able to repeat what you did (they don't know what you used). Throughout, you need to ensure that the reader knows the (relevant background) to the study area, exactly the data you used, and then what you did with it. Reply: Thank you very much. We will add the following information: The details of the datasets are shown in the following table. The area of Old Beichuan County is 2.66Km2. MS 8.0 Wenchuan earthquake caused Beichuan County massive destruction,and resulted in casualties more than ten thousand.The whole Beichuan County has become in ruins.Field seismic investigation showed that the three principal causes of such huge destruction are as follows:(1)the vibration failure effect caused by macroseism vibration;(2) the earth's surface rupture effect caused by seismo-active fault slippage;and(3) the secondary geological hazards(collapse,landslide and debris flow) caused by the earthquake. The information of the datasets used in the experiments Datasets Acquisition time Spatial resolution UAV optical multispectral image 2013.10 0.25m Ground-based LiDAR 2013.10 0.5m SAR (TerraSAR) 2014.5 1m

6.Section 3. Seismic Characteristics of multi-source remote sensing images. This is fairly description rather than quantitative in its presentation of the seismic characteristics one can detect using remote sensing images. There are some good parts in here, but can the section be made slightly more organized in its structure. This is evident also in having just one reference cited for the entire section. Thas there really been no one else who has looked at seismic characteristics using multi-source remote sensing images? Reply: Thank you. Your suggestion is right. We will adjust the structure of the chapters to make them more logical, and supplement the corresponding references. Sorry about it. We have done the analysis of seismic damage characteristics by ourselves, so we have not quoted other references, but we are sorry that we have not

quoted references in the feature calculation. We will supplement and revise them in the revised manuscript. 7.Section 4. Methodology. In terms of structure, this borders on narrative in places and coule be slightly better organized in terms of "We did the following steps: (i) ****, (ii) ****, (iii) *****" with any appropriate references. In terms of content of the methodology, although parts of this are good, imagine someone who does not have your work, trying to now read it and replicate it. Have you put in enough details for that person to reproduce each step. So give this to one of your (student) colleagues NOT familiar with the work, and ask them if they could reproduce each step over an hour. Reply: Sorry about it. In the revised manuscript, we will add the following contents. We did the following steps: (i) Data processing, including optical image, SAR image, DSM image, and building sample distribution , (ii) Feature selection , including the spectral features of optical images, the texture features of SAR images and the geometric features of DSM images (iii) Logistic regression Model construction and analysis. At the same time, in the introduction of each step, the software, process and detailed setting parameters of the method will be supplemented. 8.Section 5 and 6. I'd like to better understand the behaviour of your results and the uncertainty. So in practice, what would it mean if we were to use your algorithm in another region? Would we get 50Reply: Thanks for your advices. The main purpose of this paper is to construct a method to extract earthquake damage information. Therefore, if this method is used in other areas, only the data meet the conditions, of course, if the data type is not complete, this method can also be used. We will supplement the correct recognition rate and wrong classification rate of earthquake damaged buildings in the revised manuscript, so that we can directly see the accuracy of information identification of this method. Your suggestion is very right. The input content of the method proposed in this paper is mainly image features. When the input features are more comprehensive, the seismic damage information extracted by fitting equation will be more accurate. If the image quality is not very good (for example, there are clouds in the image, the spatial resolution of the image is low), these will also affect the accuracy of information recognition. We will also

supplement the discussion in the revised version. 9.Equations. Equation 1. Infoterra is an 'interesting' source. Do you genuinely have no other sources of reference for this key equation? What is sigma standing for âËŸA ËĞT you have not told us, nor why it is raised to the 0th power (I guess this is your radar sigma naught value). What is s in the $k_s$. $Some problems for this and the other equations with formatting. Tell us the range of values here, and why DN is the absolute val\ldots$

$Sorry about it. The references is as follows : Infoterra, Radiometric Calibration of TerraSAR-$

$X Data, TSXX - ITD - TN - 0049, 2008. The reference is TerraSAR data product manual. The calculation method is also cited h\ldots$

$sectional area (i.e. backscattering coefficient) per unit area, which can be used to characterize the scattering ability of the target t\ldots$

$Sorry about it. In the revised manuscript, we will standardize and unify abbreviations. 11. Variables. You seem to go back and fort\ldots$

$600''[?m] A'c Equation in Section 5.2 Reply : Sorry about it. We will check and revise the units of variables. 12. T give this a number,$

$BR) + (0.419 * CON''. Remind us what the acronyms mean. Reply :$

$Sorry about it. We will add relevant contents 13. Conclusions. I'm not convinced whether this is a paper that really is a new method. T\ldots$

$The method proposed in this manuscript is a new method in the field of earthquake damage information identification. In the con\ldots$

$***, by ****). A'c Figure 6. Variables go back and forth between p and P. Figure caption needs to be more complete. The colours made\ldots$

$8. Define what you mean by ME, V A, HOM, DI, etc., in the figure caption. Why different colours. Poorly done labels in places A\ldots$

$Sorry about it. We will unify the standard for the caption and legend of all figures in the manuscript. Especially for the problems r\ldots$

$Sorry about it. We will unify the standard of all tables in the manuscript.$